# Rates and Determinants of Hospital-Acquired Infection among ICU Patients Undergoing Cardiac Surgery in Developing Countries: Results from EMERGENCY’NGO’s Hospital in Sudan

**DOI:** 10.3390/antibiotics11091227

**Published:** 2022-09-09

**Authors:** Ornella Spagnolello, Silvia Fabris, Gina Portella, Dimiana Raafat Shafig Saber, Elena Giovanella, Manahel Badr Saad, Martin Langer, Massimo Ciccozzi, Gabriella d’Ettorre, Giancarlo Ceccarelli

**Affiliations:** 1Intensive Care Unit, Salam Centre for Cardiac Surgery, EMERGENCY’NGO, Khartoum, Sudan; 2Department of Public Health and Infectious Diseases, University of Rome “La Sapienza”, 00185 Rome, Italy; 3National Centre for Control and Emergency Against Animal Diseases and Central Crisis Unit—Unit III, Directorate General for Animal Health and Veterinary Medicinal Products, Italian Ministry of Health, 00153 Rome, Italy; 4Medical Statistics and Epidemiology Unit, Campus Bio-Medico University, 00128 Rome, Italy; 5Migrant and Global Health Research Organization (Mi-HeRO), Rome 00176, Italy

**Keywords:** cardiac surgery, intensive care unit, third-world countries, Sudan, infection prevalence, antimicrobial resistance, epidemiology, *Salam* Centre for Cardiac Surgery, EMERGENCY NGO, rheumatic heart disease

## Abstract

Introduction. Knowledge of local and regional antimicrobial resistance (AMR) is crucial in clinical decision-making, especially with critically ill patients. The aim of this study was to investigate the rate and pattern of infections in valvular heart disease patients admitted to the intensive care unit (ICU) at the *Salam* Centre for Cardiac Surgery in Khartoum, Sudan (run by EMERGENCY NGO). Methods. This is a retrospective, observational study from a single, large international referral centre (part of a Regional Programme), which enrolled patients admitted to the ICU between 1 January and 31 December 2019. Data collected for each patient included demographic data, operating theatre/ICU data and microbiological cultures. Results. Over the study period, 611 patients were enrolled (elective surgery *n* = 491, urgent surgery *n* = 34 and urgent medical care *n* = 86). The infection rate was 14.2% and turned out to be higher in medical than in surgical patients (25.6% vs. 12.4%; *p* = 0.002; OR = 2.43) and higher in those undergoing urgent surgery than those undergoing elective (29.4% vs. 11.2%; *p* = 0.004; OR = 3.3). Infection was related to (a) SOFA score (*p* < 0.001), (b) ICU length of stay (*p* < 0.001) and (c) days from ICU admission to OT (*p* = 0.003). A significant relationship between the type of admission (elective, urgent surgery or medical) and the presence of infections was found (*p* < 0.001). The mortality rate was higher among infected patients (infected vs. infection-free: 10.3% vs. 2.1%; *p* < 0.001; OR = 5.38; 95% CI: 2.16–13.4; *p* < 0.001). Conclusions. Hospital-acquired infections remain a relevant preventable cause of mortality in our particular population.

## 1. Introduction

Patients in the cardiac surgery intensive care unit (ICU) are particularly vulnerable to post-operative infection, considering the underlying chronic disease, prolonged and complex surgical techniques used and exposure to life-saving invasive procedures [1]. Although many studies have focused on the topic, scientific papers documenting the extent of ICU-acquired infection in low-income areas are sparse.

Particularly in East African countries, where surveillance capacity is minimal, reducing the impact of infections remains an unsolved challenge. Moreover, despite a wide range of risk factors contributing to the occurrence of infections in critically ill patients that have been described in developed countries [2,3,4,5], to the best of our knowledge, little information is available on this topic in cardiac surgery ICUs in low-income settings.

The aim of our study is to assess the epidemiology and clinical impact of infections, together with their risk factors, in patients admitted to the ICU of an international referral centre for cardiac surgery in Sudan.

## 2. Methods

### 2.1. Study Design, Patient Selection and End Points

This is a retrospective observational study carried out from 1 January to 31 December 2019, which enrolled patients admitted to a cardiac surgery ICU in East Africa (Khartoum, Sudan). In the given study period, for each consecutive patient admitted to the ICU, a study sheet was filled in when the patient was discharged. Patients re-admitted to the ICU were excluded from our analysis.

Patients were divided on the basis of the different types of hospital admission:(1)*Elective*: For clinically stable patients scheduled for cardiac surgery. Ideally, these patients are meant to be admitted on a ward level a few days before undergoing surgery in order to perform a standard pre-operative assessment including laboratory exams, ECG and heart ultrasound. Following surgery, they are initially referred to the ICU and then to the sub-ICU/ward for post-operative care.(2)*Urgent*: For patients requiring urgent care as a result of a precipitating condition. These patients receive medical care and, when appropriate, some of them also undergo urgent surgery. In relation to their clinical conditions, at hospital admission, patients are referred either to the ICU, sub-ICU or ward.

Therefore, patients were finally grouped as (1) elective surgery, (2) urgent surgery or (3) urgent medical care.

The primary outcome was to evaluate the rate of hospital-acquired infection in the ICU.

Secondary outcomes were the identification of (1) pathogens and site of infection, (2) risk factors associated with infection occurrence and (3) ICU crude mortality rate and ICU infection-associated mortality rate.

Moreover, we compared medical and surgical patients, and, in the latter group, we compared patients undergoing elective surgery with those treated urgently.

### 2.2. Study Setting

The *Salam* Centre for Cardiac Surgery in Khartoum, Sudan, is a cardiac hospital run by an Italian NGO (EMERGENCY NGO) that provides high-quality care for child and adult patients with underlying heart conditions [6]. Its services are entirely free of charge and provided not only to locals but also patients coming from other parts of Sudan, other African countries and even Iraq and Afghanistan (through the Regional Programme).

The facility includes a surgical block with three operating theatres (OTs), a 15-bed intensive care unit (ICU), a 16-bed sub-ICU and a 32-bed ward.

Patients admitted to the centre are mainly suffering from valvular heart disease (VHD) (most often rheumatic heart disease, RHD) or congenital heart disease (CHD).

Most of the procedures performed at our centre take the form of open-heart surgery with cardiopulmonary bypass.

### 2.3. Definitions

Infection severity was determined using a sequential organ failure assessment (SOFA) score in the first 24 h from infection onset.

The length of hospital and ICU stay were calculated as the number of days from the date of admission to the date of discharge or death. Re-admissions to the ICU were ruled out.

Patients were divided into infected and not infected on the basis of the presence or absence of at least one infectious event (defined as at least one positive microbiological isolate together with clinical signs of infection). Patients that did not develop any infections were defined as “censored”.

Infections were classified according to the following categories: Ventilator-associated pneumonia (VAP), urinary tract infection (UTI), abdominal infections and bloodstream infection (BSI).

Multi-drug resistance (MDR) definition was in line with the Centers for Disease Control and Prevention’s guidelines.

### 2.4. Microbiological Studies

Microbial isolates were collected in cases of fever or suspicion of infection and were classified as infection or contamination in accordance with clinical judgment. The site of specimen collection was specified (respiratory tract, blood, CVC tip, urine, tissue fluids or swab).

Identification of microbial strains was based accordingly on local laboratory techniques. The Kirby–Bauer disk diffusion test was used for antimicrobial susceptibility testing.

### 2.5. Statistical Analysis

We performed an explanatory analysis of all patients separately by groups: Categorical data are presented as absolute frequencies and percentages and continuous data as the median with the minimum and maximum range and the mean with standard deviation (SD). Categorical variables were compared using the Chi-Square test, whereas continuous variables were compared using the Mann–Whitney U test for means and the Mood median test for medians. We evaluated, by explanatory analysis, frequencies of specimens positive for infection with attention to antibiotic resistance. We obtained overall bacterial prevalence in our population and observed infection prevalence by the site of the specimen, not considering bacteria found more than once at the same site. We investigated any difference between infected and non-infected patients in terms of frequencies, central tendency and variability measures of variables we believed could be associated with infection. We tested differences using the Chi-Square and Mann–Whitney U tests. Then, we evaluated risks by the odds ratio (OR). In order to define the best predictors for multiple logistic regression, we evaluated correlations among risk factors and univariate logistic regressions AIC, pseudo R^2^ and increases in OR per unit increase in risk variable regression, as well as the number of not available (NA) observations. We considered optimal pseudo R^2^ values between 0.4 and 0.2. Moreover, we explored, by a Pearson correlation test, whether significant predictors were significantly and highly or moderately correlated with each other [7]. We evaluated multi-collinearity by the variance inflation factor (VIF). Moreover, we analysed the multivariate regression model by splitting patients into training (which contain 80% of the observations) and test datasets, stratified by the presence or absence of infection. Predictive properties of the model were evaluated by sensitivity (SE), specificity (SP) and positive and negative predictive values (PPV and NPP); the optimal threshold for maximising both SE and SP was defined over the training dataset. Kaplan–Meier curves highlight differences between groups in terms of time free from infection, with attention paid to the mean time spent in the hospital without infection. Curves for days free from infection were difference-tested using the log-rank test. All test results were evaluated considering α = 0.05; normality was assessed by the Shapiro–Wilk test. Missing data were not transformed, and in order to compute specific statistics, patients who did not report the specific information were excluded.

### 2.6. Ethics

The study was carried out in accordance with the Helsinki Declaration. Ethical approval was not required because the study was based on a retrospective analysis of data collected for diagnostic and clinical purposes by the medical staff and stored in a deidentified manner. The study complies with the indications of the STROBE Statement checklist.

## 3. Results

### 3.1. Description of Study Population

During the study period, 680 cases of ICU admissions were recorded. With re-admissions having been ruled out, a total of 611 patients were enrolled in the study. Of them, 491 were post-elective surgery patients and 34 were urgent surgery patients, whereas the remaining 86 admissions were for urgent medical care. Demographic characteristics and clinical data of the study population, together with a comparison of surgical and medical groups, can be found in Table 1. A comparison of elective and urgent patients is given in Table 2.

### 3.2. Outcomes Evaluation

#### 3.2.1. Infection Prevalence

The overall infection prevalence was 14.2% and turned out to be higher in medical than in surgical patients (25.6% vs. 12.4%; *p* = 0.002; OR = 2.43; 95% CI: 1.4–4.21; *p* = 0.001). However, in comparison to patients scheduled for elective surgery, those undergoing urgent surgery had a significantly higher prevalence of infection (29.4% vs. 11.2%; *p* = 0.004; OR = 3.3; 95% CI: 1.5–7.27; *p* = 0.002).

#### 3.2.2. Microbiological and Clinical Characteristics of Infections

Out of 611 ICU patients enrolled, 14.2% (*n* = 87) had at least one positive microbiological culture. A total of 156 samples were found positive for microbial growth; the prevalence of different microbial isolates at the site of specimen collection and by different types of hospital admission are summarised in Figure 1.

Among Gram-positive isolates, out of 23 staphylococci isolated, 28.5% (*n* = 4, all *S. aureus*) were cefoxitin-resistant. Among all Gram-negative isolates where an antibiogram was available (*n* = 110), 32.73% (*n* = 36) were carbapenem-resistant. In particular, out of 11 *Acinetobacter baumannii* isolates with a complete antibiogram available, 2 (18.2%) were carbapenem-resistant, whereas the remaining were sensitive. Concerning *Pseudomonas aeruginosa*, 6 (50%) out of 12 isolates were carbapenem-resistant. Among *Klebsiella pneumoniae* isolates with a complete antibiogram (n = 27), 13 (48.15%) were carbapenem-resistant.

Among the 87 patients with one or more positive microbiological cultures, all had clinical signs of infection. Table 3 shows the frequency of microbiological isolates and the prevalence of different infections in the different groups.

#### 3.2.3. Risk Factors Associated with Development of Infections

Infection was related to (a) the degree of organ failures assessed by SOFA score (*p* < 0.001), (b) the number of days spent in the ICU (*p* < 0.001) and (c) days from ICU admission to OT (*p* = 0.003). We evaluated risks associated with these variables: Per unit increase in risk variables, the risk of infection increases by (a) 1.44 (95% CI 1.32–1.59; *p* < 0.001), (b) 1.56 (95% CI 1.41–1.71, *p* < 0.001), (c) and 1.39 (95% CI 1.02–1.89, *p* < 0.001), respectively.

Finally, we observed a significant association between the type of admission (elective, urgent surgery or medical care) and the presence of infection (*p* < 0.001): In particular, the risk in the medical group is 2.43 (95% CI: 1.4–4.21; *p* = 0.001) times higher than in surgical patients. Among the surgical group, patients treated urgently had a 3.30 (95% CI: 1.5–7.27; *p* = 0.0018) times higher risk than elective patients.

Among people who underwent surgery, reopening and delayed chest closure were significantly associated with infection (both *p* < 0.001): Those who underwent reopening and who had delayed chest closure had a significantly higher risk of infection: OR = 5.11 (95% CI: 2.11–12.37, *p* < 0.001) and OR = 10.98 (95% CI: 3.37–35.73; *p* < 0.001), respectively. A consistent difference in the average time spent in ECC between infected and non-infected was also observed (120 min vs. 94.8 min; *p* = 0.0015). Moreover, per minute of ECC, the risk of infection increases by 1.01 (95% CI: 1.005–1.015; *p* < 0.001). All clinically relevant variables were evaluated in terms of central tendencies, in which we found significant differences between infected and non-infected. We built univariate logistic regressions, the results of which are summarised in Appendix A. Pearson correlations between relevant variables are summarised in Appendix A.

Multiple logistic regression was performed. We modelled the risk of infection by three explanatory variables (days spent in the ICU, different types of admission if surgical or medical and SOFA score); OR and statistics are summarised in Table 4, while Figure 2 gives the logistic regression line with 95% CI. The ROC curve can be found in Appendix A.

The chance of being infection-free during the time spent in the hospital is shown in the Kaplan–Meier curves given in Figure 3a, together with the frequency distribution for censored patients over time in Figure 3b. The log-rank test shows significant differences between curves for days free from infection with a *p*-value of < 0.001. Censored patients had a mean time at risk of infection (correspondent to the time they spent in hospital) of 10.6 days (±9.3) in elective patients, 9.6 days (±4.7) in urgent patients and, finally, 6.8 days (±11.1) in medical patients, on average. Time spent at the hospital (Figure 3b) for non-infected patients shows a log-normal distribution just for urgent patients (*p* = 0.166), while elective and medical time–frequency distributions are not statistically log-normal, with *p*-values of <0.001. Infected patients spent, on average, 17.7 days at the hospital (±14.6); more specifically, elective patients spent 18 days (±11.2), medical patients spent 14.6 days (±11.6) and urgent patients spent 22.5 days (±30.5). Time spent at the hospital shows a log-normal distribution for elective and medical patients (*p*-value = 0.858 and 0.135), while days of hospitalisation are not log-normal for urgent patients (0.041).

#### 3.2.4. Mortality

The ICU crude mortality rate was 3.3%. This was significantly higher in infected patients (infected vs. infection-free: 10.3% vs. 2.1%; *p* < 0.001; OR = 5.38; 95% CI: 2.16–13.4; *p* < 0.001). The ICU crude mortality rate was higher in medical than in surgical patients (13.9% vs. 1.5%; *p* < 0.001; OR = 10.4; 95% CI: 4.146–26.486; *p* < 0.001). Although not significant, the ICU crude mortality rate was higher in urgent than in elective patients (2.9% vs. 1.4%; *p* = 1).

## 4. Discussion

Severe infections and sepsis are major factors in patients’ clinical outcomes, in particular those with underlying cardiovascular diseases [8]. Investigating nosocomial infection epidemiology and predictive factors in infection is crucial in guiding clinical decisions and has an impact on morbidity and mortality, especially in settings with limited resources where complications are not always affordable.

In our study, the overall incidence of infection during ICU stay was 14.2%, with significant differences between medical and surgical patients (25.6% vs. 12.4%). These latest data can be explained by considering that part of the medical group consisted of an extremely fragile population affected by end-stage heart failure with heavy contraindications to cardiac surgery. Indeed, the SOFA score was remarkably higher (6 vs. 4; *p* < 0.001) in the medical than in the surgical group. Moreover, the infection rate for medical patients overlapped with that for the group undergoing urgent surgery (29.4%) and was significantly higher than for those undergoing elective surgery. Primary bloodstream infections (BSIs) are significantly more frequent in the medical group, as expected in cases of consistent microbial translocation, with this being more common in patients with end-stage cardiovascular disease [9].

On the other hand, VAP was rather more common in the surgical group. This observation is in line with what is to be expected in patients undergoing cardiac surgery, in which the respiratory dynamics are functionally impaired following sternotomy, which is consistent with previous studies [10,11,12].

The higher frequency of infectious events recorded in the group undergoing urgent surgery compared to the elective group can be explained not only by the more critical condition of these patients (as indicated by the SOFA score) but also by the higher number of devices used and longer exposure time to these medical aids (CVC, Foley) [13,14,15,16]. Moreover, surgical patients treated urgently spent more time in extracorporeal circulation (ECC) and continuous veno-venous haemofiltration (CVVH) than patients treated with elective surgery: These are two well-known risk factors for hospital-acquired infection [8,17,18,19].

Overall, logistic regression analysis showed that the risk of infection increased for each day spent in ICU, for more severe SOFA scores and in patients with more compromised underlying disease. In fact, medical patients in our cohort had a higher risk of infection due to their fragile condition, which prevented them from receiving surgery.

From a microbiological point of view, a large number of infections observed were due to pathogens included in the family of *Enterobacteriaceae*. This can be explained by considering that intestinal ischaemia reperfusion that occurs during cardiac surgery and in end-stage cardiovascular diseases will induce a systemic inflammatory reaction and may cause intestinal flora translocation [20]. Interestingly, among *Enterobacteriaceae* isolated in our cohort, 40.7% were carbapenem-resistant. Moreover, for *Acinetobacter*, which caused the majority of BSI, microbial translocation can be recognised as the main pathological mechanism. However, in contrast to what has been reported in European countries, in our study, carbapenems-resistant isolates were observed in only 25% of cases. The problem of antimicrobial resistance (AMR) represents a global public health issue that is particularly relevant in developing countries [21]. In fact, AMR has multiple and varied causes [22], namely (1) unreasonable prescription and/or self-prescription of antibiotics in an area with a high burden of infection diseases; (2) poor socioeconomic conditions; (3) weak product regulation, oversight or quality control and (4) limited capacity for microbiology testing and lack of local and national surveillance in low-income areas. This problem becomes particularly crucial in hospitalised patients, especially those that are critically ill. AMR is a major cause of morbidity and mortality in the ICU [23].

It is well reckoned that ICU mortality attributable to infectious diseases is considerably higher in developing countries (DCs) (14% in North Africa vs. 6% in high-income countries) [24]. Interestingly, the overall ICU mortality rate observed at our cardiac surgery centre was 3.3%. This figure is comparable to those recorded in high-income countries [3,25]. Taking into consideration overall challenges related to this specific setting (patients’ backgrounds, limited local resources), we found these data quite interesting, as they offer positive feedback on the high standard of care delivered at our centre. Moreover, we found a consistent difference in mortality between medical and surgical patients (12% vs. 8%; *p* < 0.001). In this case, this observation can be explained by the fact that some of the medical patients were end-stage heart failure cases urgently admitted following decompensation but without further surgical options.

Our study, conducted in a difficult setting, has several limitations, which must necessarily be taken into account when interpreting the results: First of all, the hospital had a microbiology laboratory with limited resources. Secondly, it is important to underline the unavailability of detailed information on the antibiotic treatments carried out, which made it impossible to take this aspect into account in the statistical analysis. Furthermore, the therapeutic options available for difficult-to-treat infections supported by multidrug-resistant germs were limited by the availability of the carbapenem class. Finally, our study population is unique, with the rate of patients suffering from RHD being so high in comparison to the global cardiac surgery dataset. Indeed, despite rising rates of CVD and atherosclerosis in DCs, RHD still remains one of the most common cardiovascular diseases in this setting [26,27,28]. Moreover, in contrast with the Western experience of cardiac surgery, our centre is mostly dealing with young and severely malnourished patients.

Despite those limitations, to the best of our knowledge, this is the first study to provide insight into the ICU cardiac surgery patient population, infection epidemiology and outcomes from a highly specialised hospital in a low-income country in North-East Africa, contributing to the implementation of research on this topic.

## 5. Conclusions

Cardiac surgery procedures require high-standard ICUs, which are still scarce in DCs, mainly as a result of a shortage of medical expertise and public infrastructure [29]. However, the presence of those few highly specialised facilities introduced a new “variable” in the management of critical cardiac surgery patients living in low-income areas. In fact, the availability of state-of-the-art treatments could not only have an impact on clinical outcomes, but also influence the ecology in hospitals and therefore the epidemiology of possible nosocomial infections associated with cardiac surgery in critically ill patients. However, limited data are available on the impact of high-resource centres delivering cardiac surgery in Africa, and there is a growing need for studies that provide information on the topic [30,31].

## 6. Recommendation

The availability of facilities capable of delivering high standards of care in developing countries requires a strong epidemiological surveillance: policy makers should consider this new reality in infection control programs.

## Figures and Tables

**Figure 1 antibiotics-11-01227-f001:**
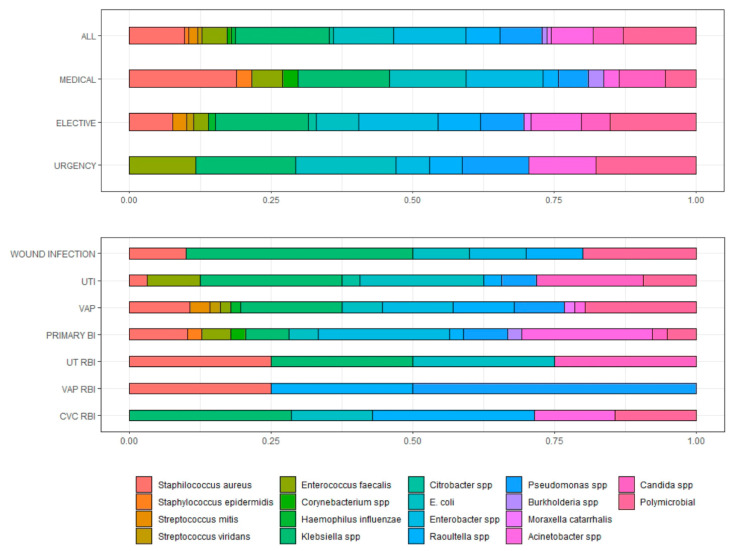
Prevalence of microbial isolates from different specimen sites and different types of hospital admission. UTI = urinary tract infection; VAP = ventilatory associated pneumonia; BSI = bloodstream infection; RBI = related blood infection.

**Figure 2 antibiotics-11-01227-f002:**
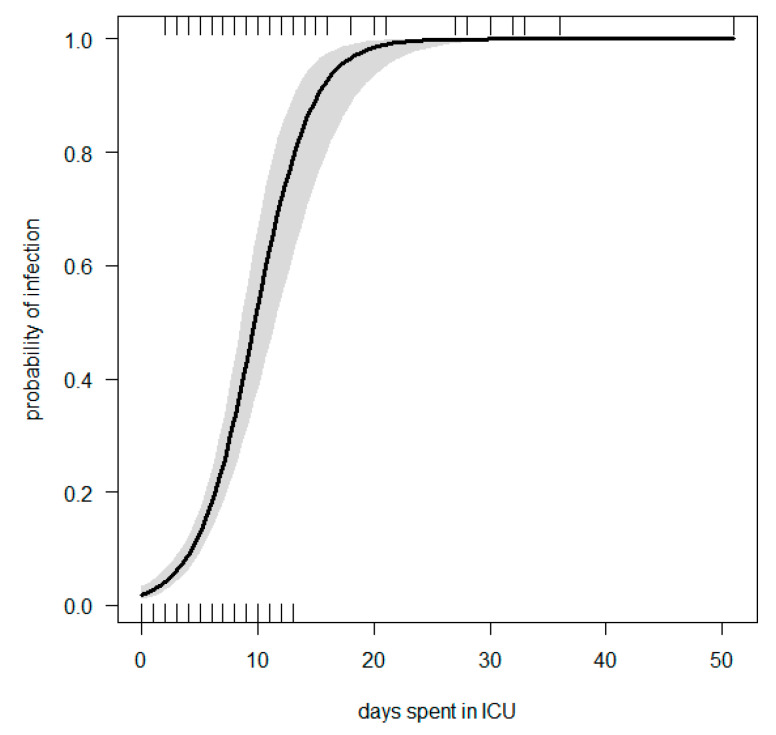
Logistic regression line: Probability of infection vs. days spent in ICU, corrected by group of admission and SOFA score.

**Figure 3 antibiotics-11-01227-f003:**
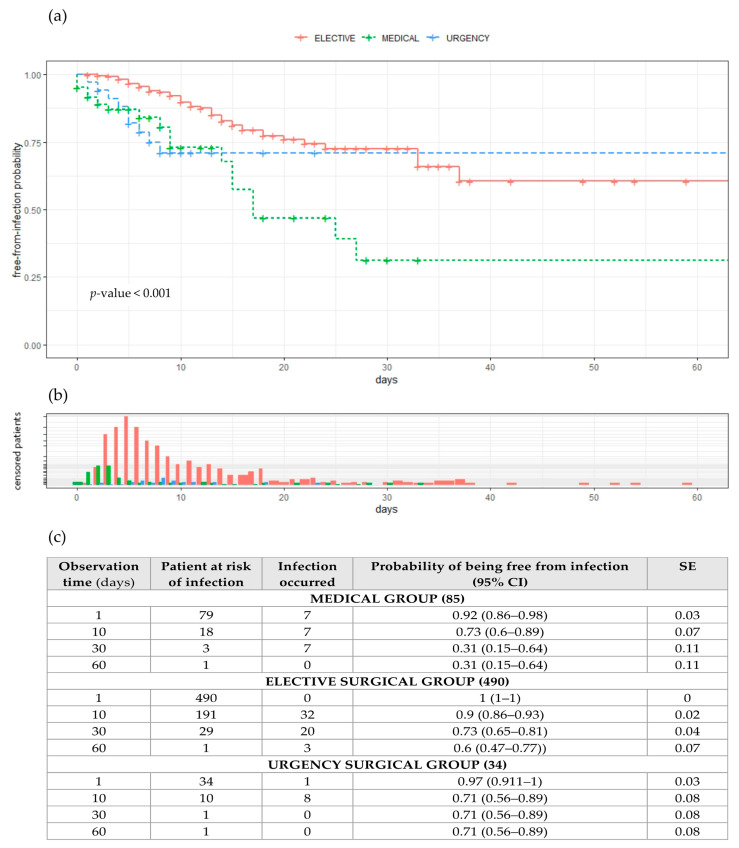
(**a**) Probabilities of being free from infection during time spent in hospital, (**b**) frequency distribution for “censored” patients over time, (**c**) observation time, patients at risk of infection, no. of infections that occurred, probability of being free from infection together with 95% confidence interval and standard error. Abbreviations: CI = confidential intervals; SE = standard error.

**Table 1 antibiotics-11-01227-t001:** Overall analysis of study population and comparison of surgical and medical groups.

	All (*n* = 611)	Medical (*n* = 86)	Surgical (*n* = 525)	*p*-Value
Age, mean (SD)	22.05 (12.6)	26.03 (13.2)	21.39 (12.45)	**0.0024**
Male, *n* (%)	299 (48.94)	40 (46.51)	259 (49.33)	0.7122
Aged less than 15 years, *n* (%)	200 (32.73)	20 (23.25)	180 (34.28)	0.05787
BMI, mean (SD)	17.86 (3.99)	18.41 (4.18)	17.77 (9.95)	0.06234
CHD, *n* (%)	497 (83.67)	3 (3.66)	94 (17.9)	**0.00146**
Patients with previous surgery, *n* (%)	10 (1.64)	1 (1.18)	9 (1.71)	1
SOFA, mean (SD)	4.7 (2.7)	6.06 (3.63)	4.49 (2.40)	**<0.001**
Time from hospital to ICU admission (days), mean (SD)	7.26 (9.43)	4.24 (10.1)	7.76 (9.23)	**<0.001**
ICU length of stay (days), mean (SD)	3.89 (4.79)	4.53 (5.5)	3.78 (4.66)	0.0617
Reintubation, *n* (%)	16 (2.62)	2 (2.32)	14 (2.67)	1
Tracheostomy, *n* (%)	5 (0.82)	1 (1.16)	4 (0.77)	1
Foley, *n* (%)—at least one	606 (99.18)	85 (98.84)	521 (99.24)	1
n. of Foley, mean (SD)	1.03 (0.24)	1.05 (0.26)	1.02 (0.23)	0.1946
Time with Foley (days), mean (SD)	4.06 (4.83)	5.07 (5.73)	3.89 (4.65)	**0.0065**
CVC, *n* (%)—at least one	604 (98.85)	81 (94.19)	523 (99.62)	**<0.001**
n. of CVC, mean (SD)	1.14 (0.57)	1.23 (0.81)	1.13 (0.52)	0.1534
Time with CVC, mean (SD)	4.02 (5.48)	4.96 (6.55)	3.87 (5.28)	0.07185
CVVH, *n* (%)—at least one session	31 (5.07)	6 (6.98)	25 (4.76)	0.5468
Time on CVVH (days), mean (SD)	0.2 (1.25)	0.22 (0.9)	0.2 (1.30)	0.39
Blood Transfusion, *n* (%)—at least one	245 (40.1)	25 (29.07)	220 (41.9)	**0.033**

Abbreviations: SD = standard deviation; BMI = body mass index; CHD = congenital heart disease; SOFA = sequential organ failure assessment; ICU = intensive care unit; CVC = central venous catheter; CVVH = continuous veno-venous haemofiltration.

**Table 2 antibiotics-11-01227-t002:** Comparison of surgical patients: Elective vs. urgent surgery.

	Surgical (*n* = 525)		*p*-Value
	Elective Group (*n* = 491)	Urgent Group (*n* = 34)	
Age, mean (SD)	21.31 (12.5)	22.65 (11.9)	0.5334
Male, *n* (%)	244 (49.69)	15 (44.12)	1
Aged less than 15 years, *n* (%)	170 (34.62)	10 (29.41)	0.6655
BMI, mean (SD)	17.82 (3.98)	17.09 (3.43)	0.3755
CHD, *n* (%)	93 (19.37)	1 (3.12)	**0.0391**
Patients with previous surgery, *n* (%)	9 (1.83)	0 (0)	0.9099
SOFA, mean (SD)	4.33 (2.34)	6.71 (2.22)	**<0.001**
Time from Hospital to ICU admission (days), mean (SD)	7.98 (8.85)	4.59 (13.4)	**<0.001**
Time from Hospital to OT (days), mean (SD)	8.03 (8.94)	6.65 (13)	**0.03824**
ICU length of stay (days), mean (SD)	3.43 (4.33)	8.82 (6.19)	**<0.001**
Reintubation, *n* (%)—at least one	13 (2.65)	1 (2.94)	1
Tracheostomy, *n* (%)	3 (0.61)	1 (2.94)	0.6231
Foley, *n* (%)—at least one	487 (99.18)	34 (100)	1
n. of Foley, mean (SD)	1.02 (0.23)	1.09 (0.29)	**0.0177**
Time with Foley (days), mean (SD)	3.52 (4.36)	9.32 (5.29)	**<0.001**
CVC, *n* (%)—at least one	489 (99.59)	34 (100)	1
n. of CVC, mean (SD)	1.1 (0.46)	1.56 (0.99)	**<0.001**
Time with CVC, mean (SD)	3.48 (4.8)	9.47 (8.14)	**<0.001**
CVVH, *n* (%)—at least one	19 (3.87)	6 (16.67)	**0.0012**
Time on CVVH (days), mean (SD)	0.17 (1.27)	0.68 (1.65)	**<0.001**
Blood Transfusion, *n* (%)—at least one	197 (40.12)	23 (67.65)	**0.003**
Number of valves repaired, mean (SD)	1.36 (0.95)	1.79 (0.91)	**0.0084**
ECC (minutes), mean (SD)	96.61 (46.61)	116.51 (49.27)	**0.0097**
Delayed chest closure	10 (2.04)	2 (5.88)	0.391
Reopening/reintervention	21 (4.28)	2 (5.88)	0.9928

Abbreviations: SD = standard deviation; BMI = body mass index; CHD = congenital heart disease; SOFA = sequential organ failure assessment; ICU = intensive care unit; OT = operating theatre; CVC = central venous catheter; CVVH = continuous veno-venous haemofiltration; ECC = extracorporeal circulation.

**Table 3 antibiotics-11-01227-t003:** Microbiological isolate and type of infections in different groups.

	All	Medical Group	Surgical Group		*p*-Value
Total number of infected patients, *n* (%)	87 (14.2)	22 (25.6)	65 (12.4)		0.004
Median number of isolates per patient (min–max)	1 (1–9)	1 (1–8)	1 (1–9)		1
Patient with one isolate (%)	53 (60.9)	13 (59.1)	40 (61.5)		1
Number of infected patients with at least one MDR, *n* (%)	32 (36.8)	9 (40.9)	23 (35.4)		0.8347
Total of isolates and rate for patient	156 (1.8)	41 (1.9)	115 (1.8)		0.8361
**Types of infections**					
Catheter-associated urinary tract infection	34 (21.8)	11 (26.8)	23 (20)		0.4908
Wound infection (surgical site)	10 (6.4)	5 (12.2)	5 (4.35)		0.1645
VAP	57 (36.5)	6 (14.6)	51 (44.3)		**0.0013**
Primary bloodstream infection	39 (25)	16 (39)	23 (20)		**0.0274**
CVC-related bloodstream infection	7 (4.49)	1 (2.44)	6 (5.22)		0.7653
Urinary tract-related bloodstream infection	5 (3.21)	2 (4.88)	3 (2.61)		0.8478
VAP-related bloodstream infection	4 (2.56)	0	4 (3.48)		0.5258
Infection due to MDR isolates, *n* (%) *	48 (30.8)	15 (36.6)	33 (28.7)		0.4576
			**Elective**	**Urgent**	***p*-Value**
Total number of infected patients, *n* (%)	-	-	55 (11.2)	10 (29.4)	0.0044
Median number of isolates per patient (min–max)			1 (1–9)	2 (1–4)	0.1651
One isolate (%)			36 (65.4)	4 (40)	0.2425
Number of infected patients with at least one MDR, *n* (%)			20 (36.4)	3 (30)	0.7789
Total of isolates and rate per patient			94 (1.7)	21 (2.1)	0.1193
**Types of Infections**					
Catheter-associated urinary tract infection	-	-	20 (21.3)	3 (14.3)	0.6727
Wound infection (surgical site)			3 (3.2)	2 (9.5)	0.4872
VAP			42 (44.7)	9 (42.9)	1
Primary bloodstream infection			20 (21.3)	3 (14.3)	0.6727
CVC-related bloodstream infection			4 (4.3)	2 (9.5)	0.6608
Urinary tract-related bloodstream infection			3 (3.2)	0	0.9423
VAP-related bloodstream infection			2 (2.1)	2 (9.5)	0.3107
Infection due to MDR isolates, *n* (%) *			28 (29.8)	5 (23.8)	0.7789

Abbreviations: MDR = multi-drug-resistant; UTI = urinary tract infection; HAP = hospital acquired pneumonia; VAP = ventilatory associated pneumonia; BSI = blood stream infection; * evaluated on the total of isolates.

**Table 4 antibiotics-11-01227-t004:** Multiple logistic regression on risk of infection in the ICU.

	OR (95% CI)	*p*-Value	VIF
No. of days spent in ICU	1.5 (1.36–1.65)	0.0302	1.06
Medical group compared to surgical group	2.23 (1.08–4.61)	<0.001	1.07
SOFA	1.18 (1.05–1.31)	0.002	1.08
pseudo R^2^ = 0.37		AIC = 321.24	

Abbreviations: OR = odds ratio; CI = confidential intervals; AIC = Akaike information criterion; VIF = variance inflation factor; ICU = intensive care unit; SOFA = sequential organ failure assessment.

## Data Availability

Data will be shared upon request.

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
