# Peer review of "Rates and Determinants of Hospital-Acquired Infection among ICU Patients Undergoing Cardiac Surgery in Developing Countries: Results from EMERGENCY’NGO’s Hospital in Sudan"

_antibiotics, 2022, doi:10.3390/antibiotics11091227_

Round 1
Reviewer 1 Report
The present manuscript describes pattern and rate of infection in valvular disease patients in the setting of a low income country. The work of the authors it is of particular value considering the difficulties of performing research activity with limited resources and considering the paucty of data for such a public health issue as hospital-acquired infections.
I have minor comments for the authors:
title:
ICU-->consider if you would like to keep it ICU or if you would prefere to expand in "intensive care"
ABSTRACT
Line 19 “cardio”àcardiac? (please check)
Line 27 “time in ICU”àICU length of stay
Line 31 particularàpatients
It is not clear in the aim that you are investigating hospital acquired infections. This becomes clear only at the end of the Abstract, in the conclusion sentence. I would suggest to better clarify since the beginning that the topic of your study is hospital acquired infections
Introduction
Line 55 “cardio”àcardiac? (please check)
“In the given study period, for each patient admitted to ICU a study sheet was filled in and updated on a daily basis till patient discharge from ICU.”à this is not very clear.
I would better specify:
1)if all the patients admitted were consecutively enrolled, to better define patient selection and to rule out the risk of selection bias
2)the timeframe of data collection (IE if since ICU admission till ICU discharge)
3)if there was a dedicated case report form (and or an electronic case report form), and if present consider if you would like to share the case report form as supplemental material
METHODS
Elective patients: why they should be admitted few days before surgery? Since the hospital stay increases the risk of hospital-acquired infections, was this a need related to preoperative assessment and optimization? Related to organizative needs? Please explain
Line 61: afterwards means “after the surgery”? please clarify
Line 69-70: secondary outcome: identification of pathogens and infectionsàwhat does it means "and infections"? Is the infection rate (as the primary outcome) or the infection site, so it is a different outcome compared to the primary? Please clarify
Paragraph 2.2àto better describe the dimension of the VHD/RHD in this setting, in order to allow the reader to better understand the dimension of the problem, I would suggest to add one or two sentences explaining the relevance of these medical conditions in that area
2.3 definitions: consider if it is worth to add the definitions of UTI/VAP/BSI within a supplemental material or an appendix
2.5 how did you deal with missing data? If no assumptions were made please add.
I would appreciate if the authors could describe the availability of mechanical circulatory support in the centre (IE ECMO, IABP; Impella, VAD…), as well as -if available- the type of surgery performed (interesting for readers involved in surgical clinical practice)
PAGE 9, above “3.2.4 Mortality”àplease add the number of table and the abbreviations (CI, SE) and better describe the box “patient at risk”, does it means “at risk for infection”?
DISCUSSION
Line 250-251: “This latest data, 250 apparently in contrast to what expected, can be explained considering that part of the 251 medical group consisted of an extremely fragile population affected by end-stage heart 252 failure and with heavy contraindications to cardio-surgery”. Since the data you report are in contrast with what expected, I wonder if you had any previous literature reference of other cardiac centers in order to set the benchmark for this outcome (as described for VAP few lines later).
Line 287à”ATB”, please expand the abbreviation
Author Response
Manuscript ID: antibiotics-1868887
Title: RATES AND DETERMINANTS OF HOSPITAL-ACQUIRED INFECTIONS AMONG ICU PATIENTS UNDERGOING CARDIAC SURGERY IN DEVELOPING COUNTRIES: RESULTS FROM THE EMERGENCY'NGO HOSPITAL IN SUDAN
We sincerely thank the reviewers for their suggestions aiming to improve the quality of this paper. Indeed, we have been pleased to receive their positive feedbacks and appreciation on our piece of work.
REVIEWER#1
The present manuscript describes pattern and rate of infection in valvular disease patients in the setting of a low-income country. The work of the authors it is of particular value considering the difficulties of performing research activity with limited resources and considering the paucity of data for such a public health issue as hospital-acquired infections.
I have minor comments for the authors:
TITLE
ICU-->consider if you would like to keep it ICU or if you would prefer to expand in "intensive care"
Re: We thank the reviewer#1, as suggested we changed ICU in Intensive Care Unit
ABSTRACT
Line 19 “cardio”àcardiac? (please check)
Re: We thank the reviewer for this suggestion. We have now edited “cardio” in “cardiac”.
Line 27 “time in ICU”àICU length of stay
Re: We thank the reviewer for this suggestion. We have now replaced “time in ICU” with “ICU length of stay”.
Line 31 particularàpatients
It is not clear in the aim that you are investigating hospital acquired infections. This becomes clear only at the end of the Abstract, in the conclusion sentence. I would suggest to better clarify since the beginning that the topic of your study is hospital acquired infections
Re: We thank the reviewer, as suggested we clarified this issue in the text
Introduction
Line 55 “cardio”àcardiac? (please check)
Re: We thank the reviewer for this suggestion. We have now edited “cardio” in “cardiac”.
“In the given study period, for each patient admitted to ICU a study sheet was filled in and updated on a daily basis till patient discharge from ICU.”à this is not very clear.
I would better specify:
1)if all the patients admitted were consecutively enrolled, to better define patient selection and to rule out the risk of selection bias
2)the timeframe of data collection (IE if since ICU admission till ICU discharge)
3)if there was a dedicated case report form (and or an electronic case report form), and if present consider if you would like to share the case report form as supplemental material
Re: We thank the reviewer for this suggestion. We better specified the enrollment criteria in the methods section. The study sheet will be available upon editor request.
METHODS
Elective patients: why they should be admitted few days before surgery? Since the hospital stay increases the risk of hospital-acquired infections, was this a need related to preoperative assessment and optimization? Related to organizative needs? Please explain
Re: Generally, our elective patients are admitted the day before the surgery. This allows us to run some standard exams (labs, ECG, US) and optimize the surgery, the anesthesia and the perfusion plans. The reason why this is performed as in-patients and not as out-patients is related by the fact that surgeries are normally carried out early in the morning and our patient’s population in Africa must be facilitated in their movements. What can happen is that, the surgery list can be slightly delayed for incoming emergencies or for difficult/complicated surgeries in OT. Therefore, elective patients can undergo surgery with a little of delay, which anyway is minimized as much as possible.
As suggested, we have now better addressed this part in the methods.
Line 61: afterwards means “after the surgery”? please clarify
Re: We thank the reviewer for this suggestion. We have now edited “afterwards” with “after surgery”.
Line 69-70: secondary outcome: identification of pathogens and infectionsàwhat does it means "and infections"? Is the infection rate (as the primary outcome) or the infection site, so it is a different outcome compared to the primary? Please clarify
Re: We thank the reviewer for this suggestion. We have now specified site of infection as secondary outcomes.
Paragraph 2.2àto better describe the dimension of the VHD/RHD in this setting, in order to allow the reader to better understand the dimension of the problem, I would suggest to add one or two sentences explaining the relevance of these medical conditions in that area
Re: We thank the reviewer for this suggestion. The relevance of this condition is better expanded in the discussion with related references list (lines 337-341). However, we had to keep it short considering the word count limit of the journal.
2.3 definitions: consider if it is worth to add the definitions of UTI/VAP/BSI within a supplemental material or an appendix
Re: We thank the reviewer for this suggestion. We have now entered the above definitions in the supplemental file.
2.5 how did you deal with missing data? If no assumptions were made please add.
Re: We did not transform missing data: where the information was not fully available, we computed statistics excluding patients who do not report the specific information, the number of missing data were evaluated together with other parameters to find the best predictors.
We have now specified this point in the manuscript: “Missing data were not transformed and in order to compute specific statistics, patients who do not reported the specific information were excluded.”
I would appreciate if the authors could describe the availability of mechanical circulatory support in the centre (IE ECMO, IABP; Impella, VAD…), as well as -if available- the type of surgery performed (interesting for readers involved in surgical clinical practice)
Re: We thank the reviewer for rising this point. We specified in the methods section that the procedures performed in our Centre are carried our as open heart surgery with cardiopulmonary bypass.
PAGE 9, above “3.2.4 Mortality”àplease add the number of table and the abbreviations (CI, SE) and better describe the box “patient at risk”, does it means “at risk for infection”?
Re: We thank the reviewer for this suggestion. We have now named Table n.5 and entered the related abbreviations. We have specified “patients at risk for infection”.
DISCUSSION
Line 250-251: “This latest data, 250 apparently in contrast to what expected, can be explained considering that part of the 251 medical group consisted of an extremely fragile population affected by end-stage heart 252 failure and with heavy contraindications to cardio-surgery”. Since the data you report are in contrast with what expected, I wonder if you had any previous literature reference of other cardiac centers in order to set the benchmark for this outcome (as described for VAP few lines later).
Re: We thank the reviewer for rising this point. In the literature there are no clear comparisons between the infectious risk of cardiac surgery patients undergoing or not undergoing surgery. However, our consideration arises from the fact that the infectious risk in patients undergoing surgery could potentially be implemented by the surgical act and by the postoperative period. In fact, previous works have shown that patients undergoing cardiac surgery are at high risk of health care – associated infections (HAIs), causing considerable morbidity and mortality. For this reason, we expected the infectious risk to be higher for surgical patients than for medical-only patients. (Chang BH, Hsu YJ, Rosen MA, Gurses AP, Huang S, Xie A, Speck K, Marsteller JA, Thompson DA. Reducing Three Infections Across Cardiac Surgery Programs: A Multisite Cross-Unit Collaboration. Am J Med Qual. 2020 Jan/Feb;35(1):37-45. doi: 10.1177/1062860619845494.)
To better clarify of the text we removed “apparently in contrast to what expected”
Line 287à”ATB”, please expand the abbreviation
Re: We thank the reviewer for this suggestion. We have now expanded “ABT” in “antibiotics”.
Reviewer 2 Report
I have evaluated the manuscript (Antibiotics-1868887) titled “RATES AND DETERMINANTS OF HOSPITAL-ACQUIRED 1 INFECTIONS AMONG ICU PATIENTS UNDERGOING CARDIAC SURGERY IN DEVELOPING COUNTRIES: RESULTS FROM THE EMERGENCY'NGO HOSPITAL IN SUDAN” by Ceccarelli and coworkers, and a retrospective analysis of rate and pattern of infections in valvular heart disease patients admitted to the intensive care unit (ICU) at the Salam Centre for Cardio Surgery of Khartoum, Sudan (EMERGENCY’NGO) has been done. All standard methods were used for this research. I found this article interesting for the readers and followed the journal Antibiotics’ scope.
I would recommend the article could be published in Antibiotics after minor corrections.
1. In the Supplementary, the authors need to include the tile of the article and the name of the authors.
2. In the Supplementary, Table S1 and Figure S2 need footnotes.
3. The authors could have analyzed the data for two years instead of one year to get a clearer picture.
4. The author needs to elaborate a bit in the conclusion to show the future direction of this study.
5. The authors need to follow the journal’s standard reference format available in the portal.
Author Response
Manuscript ID: antibiotics-1868887
Title: RATES AND DETERMINANTS OF HOSPITAL-ACQUIRED INFECTIONS AMONG ICU PATIENTS UNDERGOING CARDIAC SURGERY IN DEVELOPING COUNTRIES: RESULTS FROM THE EMERGENCY'NGO HOSPITAL IN SUDAN
We sincerely thank the reviewers for their suggestions aiming to improve the quality of this paper. Indeed, we have been pleased to receive their positive feedbacks and appreciation on our piece of work.
REVIEWER#2
I have evaluated the manuscript (Antibiotics-1868887) titled “RATES AND DETERMINANTS OF HOSPITAL-ACQUIRED 1 INFECTIONS AMONG ICU PATIENTS UNDERGOING CARDIAC SURGERY IN DEVELOPING COUNTRIES: RESULTS FROM THE EMERGENCY'NGO HOSPITAL IN SUDAN” by Ceccarelli and coworkers, and a retrospective analysis of rate and pattern of infections in valvular heart disease patients admitted to the intensive care unit (ICU) at the Salam Centre for Cardio Surgery of Khartoum, Sudan (EMERGENCY’NGO) has been done. All standard methods were used for this research. I found this article interesting for the readers and followed the journal Antibiotics’ scope.
I would recommend the article could be published in Antibiotics after minor corrections.
- In the Supplementary, the authors need to include the tile of the article and the name of the authors.
Re: We thank the reviewer for this suggestion. Article title and authors’ name have been added in the Supplementary.
- In the Supplementary, Table S1 and Figure S2 need footnotes.
Re: We thank the reviewer for this suggestion. Footnotes for supplementary materials have been entered.
- The authors could have analyzed the data for two years instead of one year to get a clearer picture.
Re: We thank the reviewer for this suggestion. On the grounds of the results of this pilot study, we might set up a new study with a more prorogued study period in the future.
- The author needs to elaborate a bit in the conclusion to show the future direction of this study.
Re: We thank the reviewer for this suggestion. We expanded our conclusions.
- The authors need to follow the journal’s standard reference format available in the portal.
Re: We thank the reviewer for this suggestion. We edited the reference format according with the journal request.
Reviewer 3 Report
Rates And Determinants Of Hospital-Acquired Infections Among ICU Patients Undergoing Cardiac Surgery In Developing Countries: Results From The Emergency'ngo Hospital In Sudan
Good paper
Appreciated
Minor Revision.
Please convert some areas of the result section table to figures.
Kindly add a recommendation section as a separate paragraph.
Some of your citation styles in the text are not according to the publisher's style.
Good Luck.
Author Response
- Please convert some areas of the result section table to figures.
We thank the reviewer for his suggestion. We converted Table 5 into Figure 3.
- Kindly add a recommendation section as a separate paragraph.
We thank the reviewer for his suggestion. We added the recommendation section accordingly.
- Some of your citation styles in the text are not according to the publisher's style.
We thank the reviewer for his suggestion. We worked on this point as well.
